

# A study on the classification of stylistic and formal features in English based on corpus data testing

Shuhui Li

School of Foreign Studies, South China Agricultural University, Guangzhou, Guangdong, China

## ABSTRACT

The traditional statistical and rule combination algorithm lacks the determination of the inner cohesion of words, and the N-gram algorithm does not limit the length of N, which will produce a large number of invalid word strings, consume time and reduce the efficiency of the experiment. Therefore, this article first constructs a Chinese neologism corpus, adopts improved multi-PMI, and sets a double threshold to filter new words. Branch entropy is used to calculate the probabilities between words. Finally, the N-gram algorithm is used to segment the preprocessed corpus. We use multi-word mutual information and a double mutual information threshold to identify new words and improve their recognition accuracy. Experimental results show that the algorithm proposed in this article has been improved in accuracy, recall and F measures value by 7%, 3% and 5% respectively, which can promote the sharing of language information resources so that people can intuitively and accurately obtain language information services from the internet.

## INTRODUCTION

With the promotion of the "New Silk Road Economic Belt" and "21st Century Maritime Silk Road", the Belt and Road (B&R) strategy covers 65 countries along the routes in Central Asia, South Asia, Central and Eastern Europe, and 53 official languages are spoken in the countries along the routes. Communication between countries, enterprises and people of different languages is becoming more and more frequent, and the realization of language information retrieval services of different languages with the help of the new generation of information technology is an essential aspect of the connectivity of the Belt and Road (*Wang & Wang, 2016*; *Li, 0000*).

The existing cross-language information retrieval is mainly based on text. However, this document retrieval method restricts the application and service of language information resources. With many language and text barriers, it is challenging to meet the specific needs of language information retrieval services using different language texts. With the rapid development of multimedia and mobile Internet technology, millions of people are collecting massive pictures through mobile device terminals at any time and place and then sharing them through social networks after attaching text content in different

Corresponding author
Shuhui Li, sophielee1980@126.com

languages. This information-sharing mode of mixed arrangement of pictures and texts makes language information on the internet not only cross-language text as the carrier. In addition, it is combined with images to express language information in rich media and multimodal ways. As one of the essential carriers of information dissemination, the digital image has become an indispensable part of people's daily life communication, which can provide a vivid and intuitive language environment and semantic expression of context for language communication.

In addition, the first thing that Chinese English attracts people's attention is the performance at the lexical level. Most researches on Chinese English neologisms mainly focus on their use (*Li & Chenghong, 2015*), and the investigation of actual data is primarily limited to examples (*Zheng & Guo, 2018*; *Fu & Cheng, 2012*). At the same time, some studies try to explain the linguistic features of Chinese English with the help of a small amount of data. However, most of these studies lack systematization and objectivity (*Yang, 2010*). The data on neologisms presented by these studies are scarce. The Chinese English Neologism database is an essential tool, and neologism recognition is one of the primary means to construct the neologisms database.

The internet has become the leading platform for language information resource sharing and language service publishing. How to calculate and use images to understand semantic similarity with corpus resources, intelligently and interactively recommend various corpus resources associated with image semantics, and construct English neologism corpus. This article aims to enable people to obtain language information directly and accurately from the internet.

## RELATED WORKS

### Corpus construction technology

Corpus is a large number of processed language materials. It has a given format and label. As a warehouse for storing and managing language data, a corpus is an ordered collection of authentic large-scale texts. Corpus has three distinct features: (1) language data in the corpus is the original resource of language communication in human productive labor; (2) It is a kind of digital resource that takes digital information as a carrier and carries human language knowledge; (3) It is a practical resource obtained through the process of corpus analysis and collation. The primary purpose of building a parallel corpus is to solve the problems of information collection, processing, sharing and communication between different dialects or languages to realize the sustainable development of language resource construction (*Chen, 2009*). The construction of multilingual parallel corpora has attracted significant attention at home and abroad, including in countries where English is the official language or the primary language and countries where English is the mother tongue, or most of the population speaks English. The main parallel corpus is "The International English Corpus", which consists of 20 parallel sub-corpora, and the United Nations Parallel Corpus, which contains cross-reference corpora in six languages.

However, these commonly used bilingual parallel corpora are mainly aimed at news, law, philosophy, religion and other specific fields, which are challenging to meet the needs

of the description of life and work scenes. They cannot directly serve the information association of different languages in cross-media information retrieval. In addition, it is more difficult to now assist the public in solving daily language communication difficulties. To achieve the goal of cross-language information retrieval, it is necessary to crawl or collect multilingual corpus samples from the internet and other authoritative language learning materials.

## Neologism recognition in corpus

Currently, there are three research methods of neologism recognition: those based on statistics, those based on rules, and the combination of regulations and statistics. Based on the statistical method, the neologisms are filtered by calculating various statistics. *Luo (2008)* and *Su & Liu (2013)* analyzed and compared the statistics suitable for microblog text characteristics and improved the branch entropy algorithm by weighted calculation, improving neologism extraction accuracy. This method has high flexibility, robust adaptability and sound portability, but it needs to train large-scale corpus and has some problems, such as data sparseness.

The rule-based method is to identify neologisms by constructing rules and matching rules on the morphology, meaning and part of speech of neologisms. *Duan & Huang (2016)* selected data sets describing plant species samples with more than 70,000 words as language materials and applied the N-gram algorithm to explore the automatic recognition of new words in professional fields. Based on Japanese word rules and onomatopoeia patterns, *Tahir et al. (2021)* proposed to add new nodes to the case framework of sentences to find the optimal path and identify the unknown neologisms in Japanese. The method has high accuracy for specific fields but shows poor portability, and the rule-building process requires a lot of human and material resources. The combination of statistics and rules can give full play to their advantages to a certain extent and effectively improve the recognition effect of neologisms. *Sasano, Kurohashi & Okumura (2014)* proposed an MBN-gram algorithm, which used statistical feature mutual information and branch entropy to expand and filter the neologism string. Finally, the dictionary was used to filter the neologism set. However, the N-gram algorithm produced a large number of word strings, which led to the reduction of experimental efficiency. *Yao, Xu & Song (2016)* proposed an unsupervised neologism recognition method. Their proposed method used the formula formed by combining word frequency, cohesion, freedom and three user-defined parameters, combined with a small number of filtering rules to identify neologisms from four large-scale corpora. However, the self-adaption of the parameter is not determined, which has certain limitations for automatic extraction.

# CONSTRUCTION OF CORPUS WITH CHINESE NEOLOGISMS

## PMI

The cohesion of words can be used to judge whether a phrase can form a term with a full meaning. PMI is a commonly used and stable statistic to express the cohesion of words.

The calculation formula of PMI is as follows:

$$PMI(x,y) = lb\frac{p(xy)}{p(x)p(y)} \tag{1}$$

Where $P(x)$ and $P(y)$ respectively represent the probability of word $x$ and word y appearing alone in the corpus, $P(xy)$ represents the probability of word $X$ and word $Y$ co-occurring in the corpus. When PMI $(x,y) = 0$, it indicates that the occurrence of words $X$ and $y$ are unrelated. When PMI $(x,y) \neq 0$, the words $x$ and $y$ appear independently. When the value of PMI $(x,y)$ is more significant, it indicates that the possibility of the word $x$ and word $y$ co-occurring is higher, and the word is more likely to become a new word.

Mutual information is effective in judging the probability of word formation of binary words, but it is necessary to divide phrases into two parts when applied to multiple terms.

$$PMI(A,B) = lb\frac{p(AB)}{p(A)p(B)} = \frac{1}{2}\left[lb\frac{p(A/B)}{p(A)} + lb\frac{p(B/A)}{2p(A)}\right]. \tag{2}$$

Formula (2) is the deformation formula of mutual information. Multivariate words can be divided into two parts, A and B. The division of A and B has various forms, such as "BRICs Leaders Xiamen declaration", which can be divided into "BRICs" and "leaders Xiamen declaration", "BRICs leaders" and "Xiamen declaration" and "BRICs leaders Xiamen" and "Declaration". Therefore, it is an effective method to take different forms of probability mean.

$$PMI(w_1 \cdots w_n) = lb\frac{p(w_1 \cdots w_n)}{avg(w_1 \cdots w_n)} \tag{3}$$

$$avg(w_1 \cdots w_n) = \frac{1}{n-1}\sum_{i=1}^{n-1} p(w_1 \cdots w_i)p(w_1 \cdots w_n) \tag{4}$$

Where, $w_1 \cdots w_n$ is multiple word string, $p(w_1 \cdots w_n)$ is the probability of a word string $w_1 \cdots w_n$ appearing in the corpus and $avg(w_1 \cdots w_n)$ is the average probability of different combinations of multiple-word strings.

In addition, when the frequency of words $AB$ and words $A$ was significantly higher than that of words $B$ Frequency, The value of PMI will be smaller. When a single average threshold is set, the neologisms that meet this feature cannot be recognized; set a single lower threshold, and invalid low-frequency words will appear in large numbers, reducing the accuracy of neologism recognition. Therefore, this algorithm uses the improved multi-PMI (Formula (3)) and sets a double threshold to filter neologisms.

## Branch entropy

To judge whether two words can form a phrase with complete meaning, in addition to the degree of cohesion within the word, the degree of freedom outside the word is also a measure. Branch entropy is a commonly used external statistic with high accuracy to express the probability of word formation. Its calculation formula is as follows:

$$HL(w) = -\sum_{wl} p(wl|w)lbp(wl|w) \tag{5}$$

$$HR(w) = -\sum_{wr} p(wr|w)\text{lb}p(wr|w) \qquad (6)$$

Where $p(wl|w)$ is the conditional probability of $wl$ given the left adjacency, and $p(wr|w)$ is the conditional probability of $wr$ given the suitable adjacency. The larger HL(w) and HR(w) are, the more kinds of adjacent words to the left of candidate phrase w, the more likely w is to become a new word. On the contrary, the smaller HL(w) and HR(w) values are, the fewer the types of words adjacent to the left of the candidate phrase W, and the smaller the possibility of w becoming a new word.

For example, in the corpus, "Xiongan New Area" is often connected with a variety of words such as "from," "the," and "that." At the same time, "A fixed "Area follows Xiongan New." "New Area" is preceded by "Xiongan." This proves that the probability of the three words "Xiongan New Area" being a new phrase is far greater than that of other related combinations like "from Xiongan New Area" and "Xiongan New Area."

**Neologism recognition algorithm**

The combination of statistics and rules is an effective method for recognizing neologisms. The flow chart of the commonly used neologism recognition algorithm is shown in Fig. 1.

The traditional combination method only focuses on the mutual information of two words. It does not extend to multi-word judgment, so it has the problem of low cohesion within terms. In addition, mutual information receives a reduction in accuracy. Given the above shortcomings, the traditional algorithm is improved by adding judgment of multi-word mutual information and setting the double threshold to recognize neologisms in the processed corpus.

Firstly, the N-gram algorithm is used to segment the preprocessed corpus, and a gram table of 1-gram ~5-gram containing word frequency is obtained. Then it is used to filter the words less than the threshold value, followed by the calculated value of mutual information in the threshold interval used to screen again. In addition, words' left and right boundaries are determined and filtered using the value of adjacent entropy. Finally, the candidate neologism set is selected using the PMI value. Because the neologisms are not included in the existing dictionaries, they are filtered by corresponding dictionaries to get the neologism set. The improved algorithm flow is shown in Fig. 2.

A candidate word set is needed to judge a phrase's internal cohesion and external freedom. N-gram algorithm can be used to segment corpus, which is independent. It does not require linguistic processing of the research corpus and does not need dictionaries and rules. Taking the English corpus as an example, set the sliding window size to N, slide one window at a time until the end of the text, and segment the text corpus. After that, each string of size N is a gram. The frequency of all grams is counted and filtered according to the preset threshold to form a qualified gram vocabulary. The N-gram algorithm is simple and easy to implement, but its efficiency is very low for processing large-scale corpus without setting N size. The analysis of the large-scale corpus in some studies (*Mei, Huang & Wei, 2016*; *Nazir, Asif & Ahmad, 2020*; *Cui, 2006*; *Haider et al., 2021*) found that each entry in Chinese corresponds to 2.33 English words and each English entry corresponds to 2.25

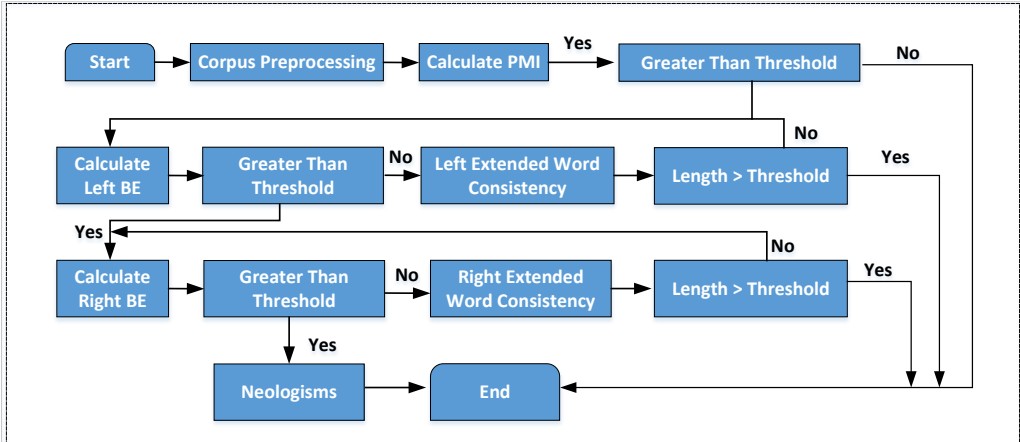

**Figure 1  The traditional neologism recognition process.** The traditional combination method only focuses on the mutual information of two words. It does not extend to multi-word judgment, so it has the problem of low cohesion within terms.

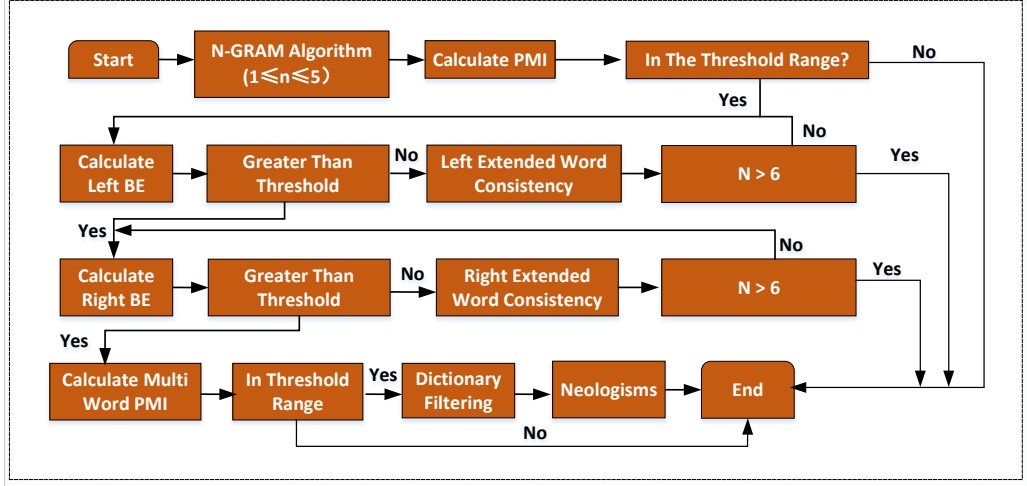

**Figure 2  The improved neologism recognition algorithm.** A candidate word set is needed to judge a phrase's internal cohesion and external freedom. N-gram algorithm can be used to segment corpus, which is independent. It does not require linguistic processing of the research corpus and does not need dictionaries and rules.

Chinese characters. Some studies have found that Chinese neologisms are mainly two to four Chinese characters. Therefore, this article uses an N-gram algorithm to process the text, setting $1 \leq N \leq 5$.

## Web crawler

Based on the neologisms recognition mentioned above, Chinese and English text data are obtained through web crawler technology, and the Chinese English parallel corpus is further constructed. As a kind of network program, a web crawler can crawl the website

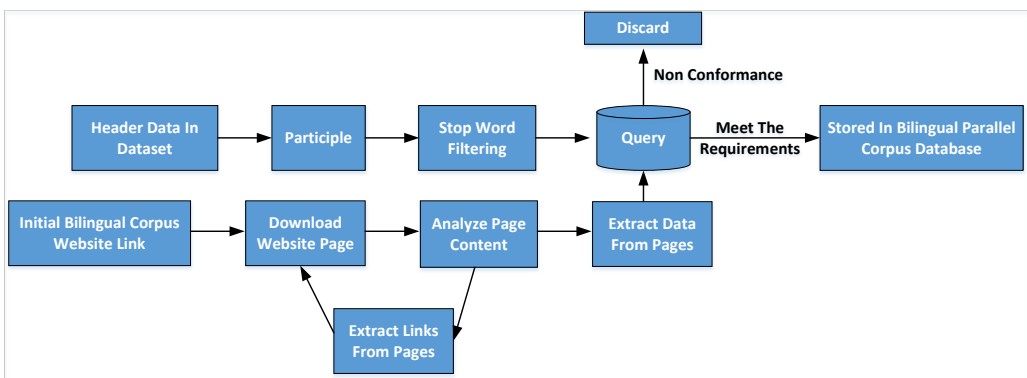

**Figure 3** **Construction process of neologisms database.** As shown in figure 3, after the initial link of the website page waiting for downloading bilingual corpus is given, the cycle execution process of the web crawler is entered.

information iteratively on the internet according to the established rules, and its basic execution process can be expressed circularly.

As shown in Fig. 3, after the initial link of the website page waiting for downloading bilingual corpus is given, the cycle execution process of the web crawler is entered. The web crawler first downloads the web page extracts the data and links from the web page and returns to the download page to continue a new cycle. The specific work contents are as follows:

(1) Download the site page

As a hypertext markup language, the web page has a fixed format. Web crawler sends commands GET one by one to the web server and requests the page according to the keyword list extracted from the dataset. After receiving the command GET, the web server returns the web page content to the web crawler client.

(2) Extract data

After downloading an HTML web page by the web crawler, the web page's content is analyzed, the text data in the web page is retained, and sentence segmentation and word segmentation are carried out. By querying the image scene vocabulary, the Chinese and English sentence pairs that meet the requirements are retained, the irrelevant HTML script statements are deleted, and the extracted Chinese English sentence pairs are stored in JSON text format.

(3) Extract links

When the network crawling task obtains data, it usually needs to get multiple pages. Because the parallel corpus data on the web page is distributed in various web pages, these Web pages have a one-to-one or one-to-many relationship with each other; that is, one Web page may contain one or more links to other Web pages. The Web pages that these links also point to have the data needed for the task. So after extracting data from a Web page, links can be extracted from that Web page and put in a queue waiting to download the page again and extract data.

## EXPERIMENT

### Corpus selection

Constructing a corpus of Chinese English neologisms includes corpus selection and corpus processing. The corpus selection mainly considers the following contents: the amount of data should be large enough; it should be widely distributed; the source should not be single; the technical means to obtain a corpus are practical and feasible. This article mainly focuses on the vocabulary of the media's official language, so the representativeness of the media should be considered in the corpus. The journals, China Daily and Shanghai Daily, were selected to assess the circulation and influence, which were founded earlier and their news was updated in time. It covers many fields, such as economy, culture, society, sports, history, etc. In the early stage of constructing the China English corpus, all English corpora of the two websites from 2012 to 2021 were obtained by purchasing authentic news corpus and using Python crawler technology.

### Corpus processing

The corpus processing work includes preprocessing the acquired corpus and recognizing neologisms in the processed corpus. The garbage string is filtered, and rules are used to remove Chinese URL links and invalid characters in the corpus. In addition, combined with the characteristics of English words, a space is used as the separator between words, which is convenient for corpus storage and neologism recognition.

### Evaluation index

The evaluation of semantic mapping results is based on @ k, that is, the proportion of the number of related results retrieved in all related results in $k$ retrieval results before output. This article uses Accuracy, Recall and F-measure to evaluate the experimental results.

$$Precision = \frac{CN}{DN} \times 100\% \tag{7}$$

$$Recall = \frac{CN}{M} \times 100\% \tag{8}$$

$$\text{F-measure} = \frac{2 \times P \times R}{DN} \times 100\% \tag{9}$$

Among them, CN is the number of phrases correctly recognized by the algorithm, DN is the number of words recognized by the algorithm, and M is the total number of terms in the corpus.

## RESULTS AND DISCUSSION

### Bidirectional retrieval

In model training, the recall rate of English-Chinese bidirectional retrieval changes with the training iteration, as shown in Fig. 4. Among them, the thick line represents the English Chinese retrieval recall rate, the thin line represents the Recall rate of Chinese English

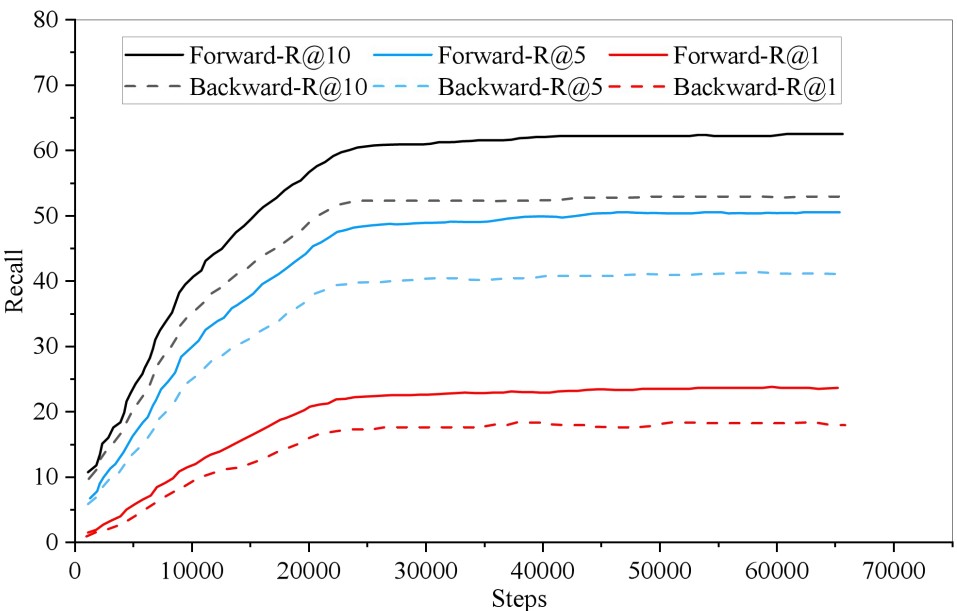

**Figure 4 Iterations of bidirectional retrieval.** In model training, the recall rate of English-Chinese bidirectional retrieval changes with the training iteration.

retrieval, and the dotted and solid lines indicate the retrieval for R@1, R@5 and R@10, respectively. As can be seen from the Fig. 5, when the Recall rate is taken as the evaluation index, the recall rate of English-Chinese bidirectional retrieval is almost the same. It can converge when it is more than 1,000 iterations.

Because the English-Chinese model is only cross-language, it has a high score in retrieval recall rate. The forward retrieval of 1,000 test data is 99.3, 99.9 and 99.9, respectively, in R@1, R@5 and R@10, while the reverse retrieval is 99.6, 99.7 and 99.9, respectively. The performance of 5,000 test data was slightly lower than that of 1,000 test data, with forward retrieval being 98.6, 99.6 and 99.7 in R@1, R@5 and R@10, and reverse retrieval being 98.3, 99.5 and 99.7, respectively.

## Neologism extraction

In this article, single threshold PMI and branch entry algorithm (ST-PMI-BE), multi-word single threshold PMI and branch entropy algorithm (MST-PMI-BE), dual-threshold PMI and branch entropy algorithm (DT-PMI-BE), and multi-word dual threshold point mutual information and branch entropy algorithm (MDT-PMI-BE) were used to test the performance of neologism extraction. The results are shown in Fig. 6.

As seen from the experimental results in Fig. 6, the proposed algorithm has a specific improvement in accuracy, recall rate and F value. Among them, multi-word mutual information is conducive to filtering multiple invalid phrases, and the setting of a double threshold can effectively retain low-frequency effective phrases and remove invalid high-frequency phrases, which proves that adding multi-word mutual information and setting a double threshold in the experiment is beneficial to the effect of neologism extraction.
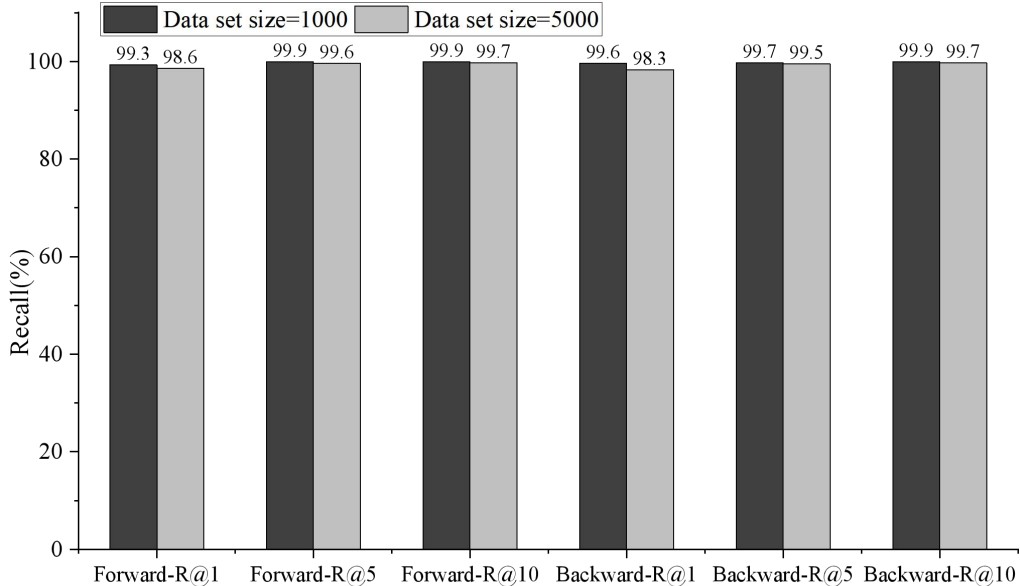

**Figure 5** **Comparison of bidirectional retrieval on different data scales.** Because the English-Chinese model is only cross-language, it has a high score in retrieval recall rate. The forward retrieval of 1,000 test data is 99.3, 99.9 and 99.9.

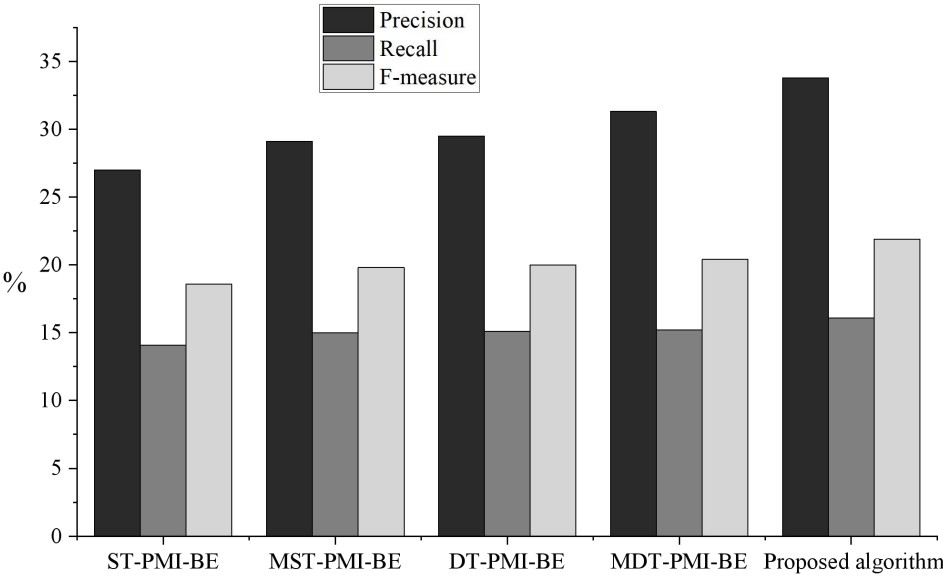

**Figure 6** **Comparison of neologisms extraction effect under different algorithms.** The results of model comparison.

Compared with the algorithm based on double threshold multi-word mutual information and branch entropy, the algorithm proposed in this article only has a higher accuracy. This is because after the corpus is segmented, there are a lot of invalid and partially overlapping phrases. Suppose the algorithm based on double threshold multi-word PMI and branch entropy is used to extract neologisms. In that case, it will not only waste the resources of the later experiment but also count these phrases, which contain some of the same words as the neologisms, into the neologism queue. Therefore, we filter these invalid words are filtered in the preprocessing stage, and the algorithm can reduce the number of invalid neologisms and improve the accuracy.

The acquired corpus is preprocessed, and the processed corpus is recognized with new words. First, the garbage string is filtered, and regular methods remove the Chinese characters, Url links and invalid characters in the corpus. Secondly, combined with the characteristics of English words, a space is used as the separator between words, which is convenient for corpus storage and new word recognition. In contrast, the adaptability of parameter values is not determined, which has some limitations for automatic extraction by computer. The portability of some methods is poor, and the rule construction process needs a lot of manpower and material resources consumption. In this article, we improve the traditional corpus preprocessing process. We add filtering for partially overlapped word strings based on removing invalid and somewhat repeated word strings. Based on conventional fusion statistics and rules method, we calculate the multi-word mutual information of the candidate word set and set a double threshold and the size of n in the n-gram algorithm to filter candidate words. The improved algorithm can not only obtain words with higher cohesion and increase the number of neologisms correctly identified but also remove a large number of meaningless word strings, reduce the number of invalid neologisms, and then recognize neologisms more accurately.

## CONCLUSION

This article mainly introduces the neologism recognition technology in constructing China's English neologisms database, which provides adequate technical support for making the corpus. This article uses the improved algorithm to recognize Chinese English neologisms, improving recognition accuracy. But on the whole, there are still some improvements in this method. It solves a large number of invalid word strings produced by traditional algorithm and improves the efficiency of calculation and matching. We should consider the misidentified words, analyze and summarize their characteristics, find the general filtering rules, and then improve the accuracy of neologism recognition. Finally, many aspects must be considered in the database of Chinese English neologisms. For example, the diversity of corpus sources, the dynamic updating of the corpus and the rationality of corpus design need further analysis and research.

### Funding

This work was supported by the 2022 Guangdong Provincial Philosophy and Social Sciences Planning Project "Research on the Teaching and Evaluation of English Writing Based on the Diagnosis of "Unilearn Platform", the project number is GD22WZX02-05. The funders had no role in study design, data collection and analysis, decision to publish, or preparation of the manuscript.

### Grant Disclosures

The following grant information was disclosed by the author:
2022 Guangdong Provincial Philosophy and Social Sciences Planning Project: GD22WZX02-05.

### Competing Interests

The author declares there are no competing interests.

### Author Contributions

- Shuhui Li conceived and designed the experiments, performed the experiments, analyzed the data, performed the computation work, prepared figures and/or tables, authored or reviewed drafts of the article, and approved the final draft.

### Data Availability

The code is available in the Supplemental Files.
The data is available at Github: https://github.com/ThunderingII/nlp_ner/tree/master/data.

### Supplemental Information

Supplemental information for this article can be found online at http://dx.doi.org/10.7717/peerj-cs.1297#supplemental-information.

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
