# Peer review of "A study on the classification of stylistic and formal features in English based on corpus data testing"

_PeerJ Computer Science, doi:10.7717/peerj-cs.1297_

## Round 0.1 · original submission · Major Revisions

Your paper needs major improvements as suggested by the experts. therefore, you are requested to carefully revise and re-submit. Please also make sure to get the language of the paper carefully edited before resubmission. Thank you.

·

Basic reporting

The traditional algorithm of combining statistics and rules lacks the judgment of the internal cohesion of words, and the N-gram algorithm does not limit the length of N, which will produce a large number of invalid word strings, which will consume time and reduce the experimental efficiency. In this paper, the improved algorithm is used to identify Chinese English new words, which improves the recognition accuracy. This paper mainly introduces the new word recognition technology in the construction of Chinese English neologism corpus, which provides effective technical support for the construction of corpus.

Experimental design

(1) Scientific and technological papers should follow the keyword indexing method of thesaurus and standardize free words as thesaurus as much as possible.

Validity of the findings

Although the results of this paper have excellent value, there are also some problems, some revisions needed to be revised to make sure that the manuscript can be accepted. The common problems are as follows:
(1) There are also some problems in language expression in this paper, which need to be modified. Please check the Chinese characters in the replacement formula and the redundant space characters in the references.

Additional comments

(1) It is suggested to modify the title for “neologism corpus” is not obvious in the manuscript.
(2) Also, the author should carefully check the statements in the paper. For example, “In addition, mutual information has an inherent disadvantage in the recognition of Chinese English neologisms that setting a single mutual information threshold will retain a large number of low-frequency meaningless phrases, resulting in the reduction of accuracy.” This statement has little relevance to the research content of the following content.
(3) The conclusion part needs to adjust the language expression, and the elaboration of this part is too lengthy. The author needs to simplify this part.
(4) Finally, it has to be added in the manuscript what kind of cognition model by other methods compared to this novel one, and it has to be outlined what is the benefit of this method.

Reviewer 2 ·

Basic reporting

Internet has now become a language information resource sharing and service the main platform, this paper studied how to calculate and use the image to understand the semantic and semantic similarity of corpus resource, intelligent and interactively recommended that is associated with the image semantic of corpus resource, and build English language corpus, enables people to intuitive and accurate language information service from the Internet.
In this paper, the improved neologism recognition algorithm is applied to the processed corpus by adding multi-word mutual information judgment and setting double thresholds. However, after corpus segmentation, the GRAM table of word frequency will change. Has the author considered this problem.
But there are still parts that need to be revised as follows:
The English of your manuscript must be improved before resubmission. We strongly suggest that you obtain assistance from a colleague who is well-versed in English or whose native language is English.

Experimental design

In this paper, the improved neologism recognition algorithm is applied to the processed corpus by adding multi-word mutual information judgment and setting double thresholds. However, after corpus segmentation, the GRAM table of word frequency will change. Has the author considered this problem.

Validity of the findings

The chart is poorly designed. The horizontal rows and vertical columns of the chart have a strict logical association, which is designed to facilitate readers to compare and compare data results, so as to intuitively get the views and conclusions conveyed by the author.
Please increase the justification of the chosen method and compare it with similar approaches in this research area. Please increase the number of references to others similar studies. A well-developed table may help to improve this part.

Additional comments

Unreasonable chart design will cause readers to misunderstand the content of the paper, cause reviewers to doubt the quality of the paper, and even deny the significance of the existence of experimental design. Authors need to double-check.
The chart is a summary and extension of the original data of the experiment, and all the data should be statistically processed in theory.
If the absence of statistical analysis means that you might only have done the experiment once, the results could be due to chance. Therefore, if statistical analysis is not available, please indicate the reason separately.

---

## Round 0.2 · accepted · Accept

The experts are agreed on accepting your article. Thanks for your interesting research contribution, and good luck with your future work.

·

Basic reporting

All changes have been incorporated as suggested

Experimental design

All changes have been incorporated as suggested

Validity of the findings

All changes have been incorporated as suggested

Additional comments

All changes have been incorporated as suggested

Reviewer 2 ·

Basic reporting

All my previous comments have been addressed. Paper qualifies my minimum criteria for acceptance and I can accept the paper.

Experimental design

Improve alot and its ok in this form now

Validity of the findings

Result have been triangulated and proper discussion have been made.

Additional comments

All my previous comments have been addressed. Paper qualifies my minimum criteria for acceptance and I can accept the paper.